# Nurses’ Influenza Vaccination and Hesitancy: A Systematic Review of Qualitative Literature

**DOI:** 10.3390/vaccines10070997

**Published:** 2022-06-22

**Authors:** Natacha Pinatel, Catherine Plotton, Bruno Pozzetto, Xavier Gocko

**Affiliations:** 1Department of General Practice, Faculty of Medicine Jacques Lisfranc, University Jean Monnet, 42000 Saint-Etienne, France; natachapinatel@hotmail.fr (N.P.); catherine.plotton@univ-st-etienne.fr (C.P.); 2Laboratory of Infectious Agents and Hygiene, University Hospital of Saint-Etienne, 42000 Saint-Etienne, France; bruno.pozzetto@univ-st-etienne.fr; 3SNA-EPIS EA4607 Laboratory, 42000 Saint-Etienne, France

**Keywords:** vaccine hesitancy, flu/influenza vaccines, qualitative research

## Abstract

Vaccine hesitancy (VH) is defined as “*delaying or refusing a secure vaccine despite its availability*”. This hesitancy affects caregivers and more specifically nurses. The purpose of this study is to assess determinants of influenza VH in the nurse’s community. We conducted a systematic review of qualitative literature according to criteria of Preferred Reporting Items for Systematic Review and Meta-Analysis and Enhancing Transparency in Reporting the synthesis of Qualitative Research from 2009 until October 2020. Eleven qualitative studies analysed (ten thematic content analyses and one grounded theory method) found three main factors in VH. The first determinant was the benefit–risk equation considered as unfavourable due to an ineffective vaccine and fears about adverse effects as the pain of the injection. Wrong immunological beliefs brought into hesitancy. Disease barriers (hand washing and masks) and personal immunity were regarded as more effective than the vaccine. Lastly, dehumanised vaccination and the difficulties of access to healthcare were institutional determinants. Nurses ask for a vaccine promotion by hierarchy and doctors with transparent information and respect for autonomy. The availability of vaccines and methods of pain control seem to be some tracks to reduce nurses’ VH.

## 1. Introduction

Vaccine hesitancy (VH) is one of the ten most important threats to public health according to the World Health Organization (WHO). WHO defines VH as « *delaying or refusing a secure vaccine despite its availability… Several factors go into this, including misinformation, complacency, convenience and trust »* [1]. In 2015, Perreti-Watel described VH as a full-fledged decision-making process: how and why people accept, refuse or postpone a vaccine. The decision-making process of VH people is unusual because this process differs from people who have full confidence and from those who reject all forms of vaccine [2]. The feeling related to vaccine safety differs across countries. In 2016, 7 of the world’s 10 most sceptical countries were in Europe. Among these, the two leading countries were France (41% of sceptics) and Bosnia-Herzegovina (36% of sceptics) versus a world average of 13% [3].

In 2017, the average number of deaths due to breathing problems associated with flu was estimated to be between 290,000 and 650,000 [4]. WHO recommends flu vaccinations in pregnant women, children between the ages of 6 to 59 months, seniors, people with chronic conditions and caregivers [5].

A European qualitative study showed that VH affected caregivers too. Caregivers focused on adverse effects of the vaccine. They did not trust the pharmaceutical industry due to a failure to share adverse effects of the vaccine and to conflicts of interest [6]. In 2017, a French qualitative study explained that the weak support for the vaccine from nurses were due to a lack of confidence in the official information sources and to an unfavourable benefit/risk balance. Nurses were doubtful about the vaccine efficiency and the collective protection (although this concept was integrated with other vaccines). Nurses in this study had difficulties promoting vaccination [7]. This same year, VH of Southern France nurses was about to 44% for overall vaccines and to 54% for the flu vaccine, which was the highest rate [8].

In France, from 2018, nurses are allowed to vaccinate adult persons to whom vaccines are recommended, including the primo injection. The only exceptions were pregnant women and patients with an ovalbumin allergy. In 2018–2019, the flu vaccination coverage in France was at 46.8% in the general population for a 75% target rate [9]. The flu vaccination coverage in health establishments was about 35% and depends on the profession (67% for doctors and midwives and 48% for nurses) [10]. In establishments providing care for the dependent elderly, the vaccine coverage was estimated at 32% (75% for doctors and 43% for nurses) [10].

Caretakers’ beliefs influence the patients’ vaccine decisions. Because nurses’ VH is higher than that of doctors, it can be a barrier to the vaccine promotion and to the vaccine coverage improvement of patients for whom vaccines are recommended, especially the most vulnerable ones [11].

The purpose of this systematic review of qualitative literature was to identify the different factors of nurses’ influenza VH.

## 2. Materials and Methods

Preferred Reporting Items for Systematic Review and Meta-Analysis (PRISMA) criteria and Enhancing Transparency in Reporting the synthesis of Qualitative Research (ENTREQ) guided the preparation and the presentation of results [12,13].

Search strategy
○Research Equation and Databases

Research was made on Pubmed and Web of Science (WOS). The research equation was developed in collaboration with a librarian, and it is described in Box 1.

Box 1.Research equation.(((((vaccine[Title] OR vaccination[Title]) AND (flu[Title] OR influenza[Title])) AND (nurses[Title] OR health[Title] OR healthcare[Title] OR nursing[Title])) AND (qualitative[Title/Abstract] OR qualitative research[Title/Abstract]) AND (barriers[Title/Abstract] OR opinions[Title/Abstract] OR motivators[Title/Abstract] OR factors[Title/Abstract] OR predictors[Title/Abstract] OR adherence[Title/Abstract] OR acceptance[Title/Abstract] OR knwoledge[Title/Abstract] OR decision-making[Title/Abstract] OR hesitancy[Title/Abstract] OR behaviours[Title/Abstract] OR attitudes[Title/Abstract] OR reasons[Title/Abstract])) NOT (children[Title/Abstract] OR pregnancy[Title/Abstract])) AND (“26 June 2009”[PDAT]: “31 December 2019”[PDAT]).

Study selection
○Inclusion criteria

To be included, the items had to talk about the nurses’ VH in their titles or their abstracts. They were to be written in English or French and were to be published between 1 January 2009 and 31 December 2019. Literature monitoring was carried out until October 2020:

Only qualitative or mixed studies were included:○Exclusion criteria.

Items that do not deal with the nurses’ flu VH in the title or the abstract were excluded. Quantitative studies, editorials and position papers were excluded.

The first selection was carried out blind by NP, CP and XG on the title and the summary. Items that do not deal with VH or nurses were excluded. The second full-reading selection of texts led to final inclusion or exclusion of items that only dealt with VH or nurses. Inconsistencies were discussed reaching consensus as the selection progressed. The flowchart (Figure 1) illustrates the assessment and selection processes of the items:Data analysis.

The complete reading of the articles allowed the extraction of the desired data and was done by NP, CP and XG.

The researchers identified two PICO (Population, Intervention, Comparator, Outcome) questions *a priori*, to examine determinants likely to influence vaccine hesitancy nurses for influenza: the benefit–risk balance and the lack of confidence in the official information.

Quality assessment of qualitative studies was based on the COREQ’ checklist (Consolidated Criteria for Reporting Qualitative Research) and carried out by NP, CP and XG (annexe 1) [14].

## 3. Results

### 3.1. Identification of Relevant Studies

Eleven qualitative studies carried out in eight different countries were selected in accordance with the previous selection criteria (Figure 1) [15,16,17,18,19,20,21,22,23,24,25].

The methods involved ten thematic analyses of the content and one grounded theory method. Concerning the data collection, nine studies used one-on-one conversations, eight of which were semi-structured and one was open. Some focus groups were used in two studies. The specific features of each study: first author, type of study, data collection, type of analysis, country, number of participants and average age are described in Table 1.

Three topics stood out throughout these different articles: an unfavourable benefit-risk balance, adherence to mistaken beliefs, and institutional determinants (lack of information and difficulties in access to healthcare).

### 3.2. Risks and the Duty to Be Healthy

Fear of some side effects of the vaccine which might be qualified as harmless was one of the determinants of VH’s nurses [15,16,17,18,19,20]. The most described symptoms were headaches, pains during the injection, and muscular pains after the injection and fever [15,16]. Only one serious side effect was mentioned, which was Guillain–Barré syndrome [16]. This fear of side effects could be explained by bad vaccine experiences or by a colleague’s experience: “…My daughter, when she had vaccinations she had a bad experience with her vaccinations, …and I fear having vaccination myself…” [15,16,18,19].

The influenza vaccine efficiency was often called into question with the main point being the annual mutations in the flu virus: “…you also don’t know when you get vaccinated against the flu—there are always different types every year—if this, exactly this type is included in the vaccine…” [15]. “I haven’t got the vaccine before. I guess I’ve got concerns about … one is the side effects of the vaccine and two is about how effective it is and, you know, whether it covers all strains of potential viral respiratory illnesses” [25]. Scientific data were considered as insufficient [17], and some recent studies had proved that the vaccine was ineffective [18,19]. Vaccines did not limit viral transmission: ‘‘I remember around 2006 when Singapore started encouraging people to have the flu vaccine. Those who had it, about half of them got flu. I didn’t take the flu vaccine, nothing happened to me…” [19].

The duty to be healthy as a result of their healthcare mission justified the fact that nurses avoided the risks involved in an ineffective flu vaccine: “…But my experience is that whenever I got the vaccination, I felt bad for a couple of months…” [18].

### 3.3. Mistaken Immunological Beliefs

The mistaken beliefs observed in nurses mostly concerned immunological data. The risk of getting flu for a nurse was considered low, and the risk of dying from flu was even lower: “…I am a healthy person and not at risk myself if I don’t get vaccinated…” [15,16,18,19,20,21,22,23]. The high personal immunity was seen as a better system of protection against flu compared to the flu vaccine itself [19,22]. A young age guaranteed this protective immunity [19-21].

The vaccine could be responsible for flu, described as being more serious than flu: *“*that had been acquired naturally*”* [19]. “…You know, it’s just gonna make you sick. You’re gonna get the flu from it, you’re gonna be ill…” [24]. “I thought it only came in one strain and not that every year it’s modified to what, like, the strain they expect for that year. I thought that people always got sick after they had the injection and not that the vaccine takes up to two weeks for it to become effective.” [25]. Some notions of resistance to the vaccine and waning immunity were also being described: “…I‘m generally questioning whether it makes sense to manipulate the immune system in such a way and to vaccinate it with anything and everything, so that it can‘t develop its own defences, right?...” [18].

The risk of transmission of flu was far widely minimized. Transmission was linked to the presence of symptoms [16,21]. The other protective measures (mask, handwashing for example) were considered more efficient than the vaccine: “…I do really good hand washing and like I said, I never get sick…” [21]. Nurses expressed their desire to protect their patients and their relatives. This desire was not correlated with the decision to vaccinate for the reason that they considered themselves as protected: “…you don’t want to pass it on to your family or anyone that you’re working with or any of your patients, especially with our patients here because most of them are quite vulnerable…” [16]. Concerning patients, influenza was neither considered as a major cause of morbidity nor as a cause of death [21,22]. Flu vaccine was not regarded as *“*being the priority*”.*

### 3.4. Institutional and Hierarchical Hesitation

The lack of official support, especially within hospital structures, was a major source of nurses’ VH [17-20]. Prevention messages were not visible enough: “I think people don’t pay attention to signs. There’s too many of them” [25]. Promotion should be done by the hierarchy and doctors. One perception of VH among doctors fostered nurses’ VH [19]. ‘‘…Maybe they also recognize that it is not really 100% proven that is why they are not pushing everyone to go for it…”. These awareness campaigns had to respect nurses’ autonomy and to give them the choice [18-20,23].

The H1N1 Influenza pandemic in 2009 strengthened a lack of confidence in the official sources of information which could be influenced by the pharmaceutical industry: “…“From my point of view all it’s really about is the money. It’s not about the patient… I think there’s a Mafia between the doctors and the pharma industry. They both benefit from each other. It’s a “lucrative” deal, it has to be produced quickly and the pharma industry makes a lot of money from it…”[18,19,22].

A gap in access to *“*vaccination centers*”* within hospital structures was also described as a VH determinant [17,19,21]. Vaccination campaigns were regarded as too short. Vaccination centers were criticized as a result of their remoteness and of a lack of vaccines. Vaccination in the workplace had to go away from a chain vaccination: *“…*compulsory, uninterrupted and dehumanized vaccination*…”.* The medical consultation had to respect autonomy and intimacy, and it had to take anxiety into account (about injection for example): “…“The other thing that has always bothered me is those campaigns that have been made. That you’re under pressure like that. That the nursing staff has to do it and that you basically have to have a bad conscience if you don’t get vaccinated…because us bad nurses will infect the patients that way…something like that…” [16,18,20].

### 3.5. Quality Assessment of the Studies

The quality of the studies based on the COREQ grid varied from 12/32 to 26/32 [18]. The least found criteria were the author features, in particular their experiences and relationships with the nurses. Triangulation with a proofreading of the interviews with the interviewees was not found in any studies. Results are more detailed in Table 2.

## 4. Discussion

### 4.1. Main Results: Nurses: A Classical VH?

Figure 2 summarizes the main results and describes the interactions between determinants of HV.

Risks associated with vaccines are common determinants to the VH caretakers and the VH parents [26]. This perception of risk to both VH parents and nurses can be based on a personal experience or that of close relatives [26]. In both populations (parents and nurses), the risk determinant is related to the benefits’ determinant. The perception of risk and benefit varies by location of work [27]. For example, a nurse who works in an intensive care unit in a teaching hospital may be able to care for a patient suffering from flu with acute respiratory distress syndrome. This experience is going to fuel their own perception of the benefits and risks of the vaccine. Thus, the most found risks of the vaccine in this review (pain on injection, fever…) can be qualified as benign by this intensive care nurse even though these risks appear as VH determinants for nurses in general and for VH parents/relatives [26]. On the other hand, a high benefit of the vaccine seems to be a major determinant of the acceptance of the vaccine [27]. In 2015, about strategies used to cope with VH, a research journal noted that none of the fifteen literature reviews or meta-analysis selected dealt with pain control during injection. This pain control was not specifically tested in people who hesitated to be vaccinated because of fear of pain as a determinant. However, tested in VH parents during their children’s inoculation, the pain control reduces anxiety about injection [28]. The perception of risk/benefit is also linked to mistaken beliefs identified in this review. The vaccine inefficiency and its risks (including a more virulent flu due to the vaccine) are enhanced by the sense that nurses will not get the disease, thinking they are protected from diseases by their job and barrier gestures. This mistaken belief can be comparable to parents’ beliefs who think that breastfeeding enables their children to get enough immunity dispensing them from vaccines [29,30].

The physicians’ knowledge is described as higher than the nurses’ knowledge [27]. This lack of knowledge is also a VH determinant among VH parents [26,29]. This review emphasised the fact that nurses need to be and feel like being informed by their hierarchy and the institutions. Dehumanized and uninterrupted vaccination without information, denying nurses’ autonomy feeds the VH process. This need for information and this desire of autonomy are also found among VH parents [26,29]. The absence of information and autonomy can lead to or strengthen a lack of scientific evidence feeling that underlies flu vaccination. The purpose of a campaign to promote flu vaccination would not only bring an increase in the number of vaccinated caretakers but also better access to information [30]. The lack of transparency from GlaxoSmithKline and from some institutions about side effects (narcolepsy) of H1N1 Pandemrix flu vaccine in 2009 accounted for VH development [31,32]. To provide the corresponding information to the VH nurses or relatives is not obvious to doctors [31]. The new criticisms of some vaccines are less radical than in the past. Historically, they came from sects linked with alternative medicines or with conspiracy theories [29]. Nowadays, controversies are more credible and have a more important place in the media. Some doctors end up doubting and the chain of command may weigh in favour of VH [31].

In the face of an upsurge of VH, some countries such as France chose to make the vaccine mandatory [31]. Certainly, these actions often showed an increase in the vaccination coverage rate, but they also strengthened the mobilisation of opponents and the loss of confidence in the institutions [31]. In this review, nurses feared these measures.

### 4.2. Strengths and Limitations

This systematic literature review allows the social and emotional aspects of the VH determinants to be revealed. VH is a complex phenomenon, and discussion, by comparing these results with the literature, showed interaction between the different determinants and their transferability to other populations. Our search strategy may have led to the omission of studies. Although the search equation was developed with the help of a librarian, it may have been too stringent. In addition, other search databases could be included such as “Embase” or “PsychINFO”. The quality of selected articles (18/32 items on average) requires a cautious interpretation of the results, even if they are comparable to the literature on the subject.

## 5. Conclusions

The main determinants of nurses’ VH are an unfavourable benefit/risk balance with a vaccine considered as ineffective and fear about adverse effects as the pain of the injection. Nurses want and need to be clearly informed in order to rectify mistaken beliefs such as the fact that they are protected by immunization through their profession or barrier gestures. Difficulty in accessing the vaccine may be a reason for vaccine hesitancy. Nurses ask for a vaccine promotion by hierarchy and doctors with a transparent information. A chain vaccination was perceived as not respecting their autonomy.

## Figures and Tables

**Figure 1 vaccines-10-00997-f001:**
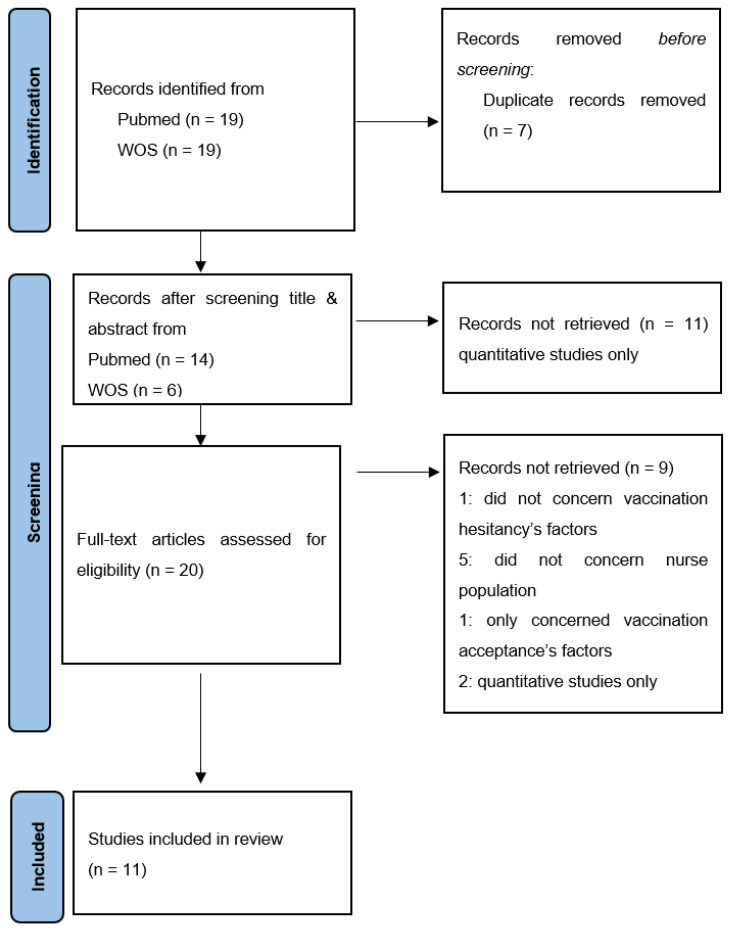
Flow diagram for systematic review: nurses’ influenza vaccination and hesitancy.

**Figure 2 vaccines-10-00997-f002:**
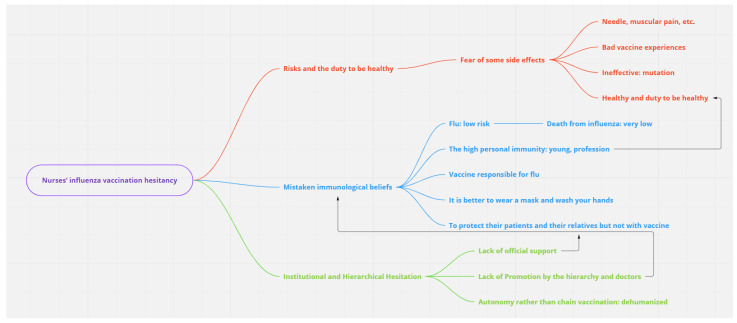
Interactions among the determinants of VH.

**Table 1 vaccines-10-00997-t001:** Study characteristics: authors, countries, number of participants, mean age, type of study, and method of qualitative analysis.

Authors Year	Country	Number of Participants	Mean Age	Type of Study	Data Collecting	Type of Qualitative Analysis
Rhudy 2010	USA	14	44 years	Qualitative	Semi-structured interviews	Thematic content analysis
Oria 2011	Kenya	113 25% of nurses	?	Mixed	Collective discussions	Thematic content analysis
Seale 2012	Australia	29 clinical nurse consultant, nurse manager	?	Qualitative	Semi-structured interviews	Thematic content analysis
Lim 2013	Australia	21	?	Qualitative	Semi-structured interviews	Thematic content analysis
Lehmann 2014	Germany, Belgium and Holland	123 46% of nurses	32–40 years	Qualitative	Semi-structured interviews	Thematic content analysis
Seale 2016	Australia	13	24–65 years	Qualitative	Semi-structured interviews	Thematic content analysis
Tuckerman June 2016	Australia	22 63% of nurses	?	Qualitative	Semi-structured interviews	Thematic content analysis
Pless November 2016	Swiss	18 100% of nurses	?	Qualitative	Semi-structured interviews	Thematic content analysis
Pless 2017	Swiss	18 100% of nurses	?	Qualitative	Semi-structured interviews	Thematic content analysis
Sundaram March 2018	Singapour	73 30% of nurses	37 years	Qualitative	Collective discussions	Thematic content analysis
Benin 2018	USA	21	?	Qualitative	Open discussions	Grounded theory

**Table 2 vaccines-10-00997-t002:** Quality assessment of selected studies with the COREQ grid.

Studies	Rhudy 2010	Oria 2011	Seale 2012	Lim 2013	Lehmann 2014	Seale 2016	Tuckerman 2016	Pless 2016	Pless 2017	Sundaram 2018	Benin 2018
		RESEARCH TEAM AND REFLEXIVITY									
**1. Interviewer**			**x**	**x**	**x**	**x**	**x**	**x**	**x**	**x**	**x**
**2. Credentials**								**x**			
**3. Occupation**			**x**		**x**	**x**	**x**	**x**		**x**	**x**
**4. Gender**			**x**			**x**	**x**	**x**			
**5. Experience**				**x**	**x**			**x**	**x**	**x**	
**6. Relationship** **established**								**x**			
**7. Participant** **knowledge** **of the interviewer**	**x**							**x**		**x**	
**8. Interviewer characteristics**			**x**			**x**					
		STUDY DESIGN									
**9. Methodological orientation and theory participant selection**	**x**	**x**	**x**	**x**	**x**	**x**	**x**	**x**	**x**	**x**	**x**
**10. Sampling**	**x**		**x**	**x**	**x**	**x**	**x**	**x**	**x**	**x**	**x**
**11. Methods of approach**	**x**		**x**	**x**	**x**	**x**		**x**	**x**	**x**	
**12. Sample size**	**x**	**x**	**x**	**x**	**x**	**x**	**x**	**x**	**x**	**x**	**x**
**13. No-participation**	**x**	**x**	**x**	**x**			**x**	**x**	**x**		
**14. Setting of data collection**	**x**			**x**	**x**		**x**	**x**	**x**	**x**	
**15. Presence of no- participants**	**x**	**x**		**x**	**x**		**x**	**x**	**x**	**x**	**x**
**16. Description of sample**	**x**	**x**		**x**	**x**	**x**	**x**	**x**	**x**	**x**	**x**
**17. Interviewer guide**	**x**	**x**	**x**	**x**	**x**	**x**	**x**	**x**		**x**	**x**
**18. Repeat interviews**	**x**			**x**	**x**		**x**	**x**	**x**	**x**	**x**
**19. Audio/video recording**	**x**	**x**	**x**	**x**		**x**	**x**	**x**	**x**	**x**	**x**
**20. Field notes**	**x**	**x**		**x**			**x**	**x**	**x**	**x**	
**21. Duration**	**x**	**x**	**x**		**x**	**x**	**x**	**x**	**x**		**x**
**22. Data saturation**					**x**	**x**	**x**	**x**	**x**	**x**	
**23. Transcripts returned**	**x**	**x**		**x**	**x**		**x**	**x**	**x**	**x**	**x**
		ANALYSIS AND FINDINGS									
**24. Number of data coders**			**x**		**x**		**x**	**x**	**x**	**x**	
**25. Description of coding tree**	**x**	**x**		**x**	**x**		**x**	**x**	**x**	**x**	**x**
**26. Derivation of themes**	**x**	**x**		**x**	**x**		**x**	**x**	**x**	**x**	**x**
**27. Software**				**x**	**x**		**x**			**x**	
**28. Participants checking reporting**											
**29. Quotations presented**	**x**	**x**		**x**	**x**		**x**	**x**	**x**	**x**	**x**
**30. Data and findings consistent**	**x**	**x**	**x**	**x**	**x**	**x**	**x**	**x**	**x**	**x**	**x**
**31. 32. Clarity of majors and minors themes**	**x**	**x**	**x**	**x**	**x**	**x**	**x**	**x**	**x**	**x**	**x**

Each x means that the study gives information required to respond to items.

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
