# Peer review of "Nurses’ Influenza Vaccination and Hesitancy: A Systematic Review of Qualitative Literature"

_vaccines, 2022, doi:10.3390/vaccines10070997_

Round 1

Reviewer 1 Report

In this manuscript, Natacha et al conducted a systematic review of the qualitative literatures from 2009 until June 2019 to assess determinants of influenza vaccine hesitancy in the nurse’s community. Authors showed three main factors in influenza vaccine hesitancy of nurses according to nine qualitative studies (eight thematic content analyses and a grounded theory method), availability of vaccines and methods of pain control were thought to be some tracks to reduce nurses’ VH. However, new findings in this review are limited, and some issues remain to be clarified.

Major comments:

  1. The research significance of this study and the reason for selecting nurses instead of caregivers as the target population remained to be clarified. The background information such as the impact of VH of nurses on the disease control of influenza should be fully presented in the Introduction part.
  2. Current search strategy may lead to the omission of relative studies. i). More searching databases should be included, such as “Pubmed”, “Embase”, “Web of Science” , in addition to “Medline” (Medline is the subset of Pubmed); ii). The research equation was too strict and only 19 studies were identified. Some suggestions are as follows: 1) vaccine[Title/Abstract] OR vaccination [Title/Abstract]; 2) flu[Title/Abstract] OR influenza[Title/Abstract]; 3) nurses[Title/Abstract] OR health[Title/Abstract] OR healthcare[Title/Abstract] OR nursing[Title/Abstract]); 4) (qualitative[All Fields] OR qualitative research[All Fields]); 5) (children[Title/Abstract] OR pregnancy[Title/Abstract]); and 6) ("2009/06/26"[PDAT] : "2019/06/23"[PDAT]). The research equation could be ((#1 and #2 and #3 and #4 and #6) not #5). iii). The secondary reference search should be conducted.
  3. “Materials and Methods”, the primary or secondary analysis objectives, the process of information extraction and the process of divergences handle should be describe

Minor comments:

  1. Several tables in the manuscript appear to have formatting issues which prevent the full table content from being completely displayed.

Author Response

In this manuscript, Natacha et al conducted a systematic review of the qualitative literatures from 2009 until June 2019 to assess determinants of influenza vaccine hesitancy in the nurse’s community. Authors showed three main factors in influenza vaccine hesitancy of nurses according to nine qualitative studies (eight thematic content analyses and a grounded theory method), availability of vaccines and methods of pain control were thought to be some tracks to reduce nurses’ VH. However, new findings in this review are limited, and some issues remain to be clarified.

Thanks for the general comment, the authors of this work were quite surprised by the results. For example, the fear of pain does not seem to us to be well known or rarely taken into consideration for the nurses' VH.

Major comments:

  1. The research significance of this study and the reason for selecting nurses instead of caregivers as the target population remained to be clarified. The background information such as the impact of VH of nurses on the disease control of influenza should be fully presented in the Introduction part.

Thanks for the comment, we are sorry that our reasoning is not clear to you and we have added a sentence.

  1. Current search strategy may lead to the omission of relative studies. i). More searching databases should be included, such as “Pubmed”, “Embase”, “Web of Science” , in addition to “Medline” (Medline is the subset of Pubmed); ii). The research equation was too strict and only 19 studies were identified. Some suggestions are as follows: 1) vaccine[Title/Abstract] OR vaccination [Title/Abstract]; 2) flu[Title/Abstract] OR influenza[Title/Abstract]; 3) nurses[Title/Abstract] OR health[Title/Abstract] OR healthcare[Title/Abstract] OR nursing[Title/Abstract]); 4) (qualitative[All Fields] OR qualitative research[All Fields]); 5) (children[Title/Abstract] OR pregnancy[Title/Abstract]); and 6) ("2009/06/26"[PDAT] : "2019/06/23"[PDAT]). The research equation could be ((#1 and #2 and #3 and #4 and #6) not #5). iii). The secondary reference search should be conducted.

Thank you for your comment and suggestion. The search equation was developed with a librarian specializing in equation development. The search was done on pubmed and we have corrected the method: thank you. Unfortunately, we are not able to take the work from the beginning. The other reviewers asked us for other important modifications such as the addition of verbatims and a scheme. We were forced to make choices within the 10-day deadline. We have incorporated your proposal into the limits of our work.

  1. “Materials and Methods”, the primary or secondary analysis objectives, the process of information extraction and the process of divergences handle should be describe. Thank you for your comment a sentence has been added.

Minor comments:

  1. Several tables in the manuscript appear to have formatting issues which prevent the full table content from being completely displayed. Thanks for your comment, we think this problem is due to the inclusion of tables in the template. We have the tables at your disposal in word format.

Reviewer 2 Report

The authors undertook a systematic review of the reasons behind vaccine hesitancy to influenza virus vaccines in nurses. It’s a very interesting topic and very relevant to the current situation about VH towards the SARS CoV2 vaccine that has been rolled out around the world, and in many countries where vaccines mandates are common.

The power of qualitative studies is that it can provide a rich depth of analysis and gives a ‘voice to the person’ being interviewed. Sadly this was missing from the systematic review that was presented.  There was no use of statements/ quotes from participants. This aspect really undersold the value of the systematic review.   You could have used key statements/ quotes to highlight the concern of VH of nurses who lacked confidence in vaccines, or their doctors.

The last sentence in the conclusion are points that should be expanded on in the discussion. If you were reviewing qualitative studies then there must be some key statement that could be highlighted from the studies reviewed. There was nothing about motivational interviews or methods of pain control.  These topics could have been explored in more detail.

Minor concerns:

Line 176 typo warning change to waning

Line 223  is going to fuel his own perception – this should read  is going to fuel their own perception…

Line 258  they came from sects linked with…  should be written they came from sectors linked with …

to avoid dehumanized and uninterrupted vaccinations – this statement was made a couple of times in the manuscript but the authors need to provide some further clarity of what exactly they are referring to here. It is not obvious form the context of the sentence in which it is used.

The last sentence in the conclusion are points that should be covered in the discussion. If you were reviewing qualitative studies then there must be some key statement that could be highlighted from the studies reviewed.

Author Response

The authors undertook a systematic review of the reasons behind vaccine hesitancy to influenza virus vaccines in nurses. It’s a very interesting topic and very relevant to the current situation about VH towards the SARS CoV2 vaccine that has been rolled out around the world, and in many countries where vaccines mandates are common. Thank you for your general comment and the interest in our work.

The power of qualitative studies is that it can provide a rich depth of analysis and gives a ‘voice to the person’ being interviewed. Sadly this was missing from the systematic review that was presented.  There was no use of statements/ quotes from participants. This aspect really undersold the value of the systematic review.   You could have used key statements/ quotes to highlight the concern of VH of nurses who lacked confidence in vaccines, or their doctors. Thank you for your comment, we had hesitated and considering your remark we have integrated verbatims.

The last sentence in the conclusion are points that should be expanded on in the discussion. If you were reviewing qualitative studies then there must be some key statement that could be highlighted from the studies reviewed. There was nothing about motivational interviews or methods of pain control.  These topics could have been explored in more detail. Thank you for your comment, in fact this sentence wanted to open on a related subject, we propose to delete it.

Minor concerns:

Line 176 typo warning change to waning thank you: corrected

Line 223  is going to fuel his own perception – this should read  is going to fuel their own perception… thank you: corrected

Line 258  they came from sects linked with…  should be written they came from sectors linked with … thank you: but we want to say sects

to avoid dehumanized and uninterrupted vaccinations – this statement was made a couple of times in the manuscript but the authors need to provide some further clarity of what exactly they are referring to here. It is not obvious form the context of the sentence in which it is used. Thank you: we have added the term chain vaccination.

The last sentence in the conclusion are points that should be covered in the discussion. If you were reviewing qualitative studies then there must be some key statement that could be highlighted from the studies reviewed. thank you: deleted

Reviewer 3 Report

The manuscript by Natacha et al. “Nurses’ influenza vaccination and hesitancy A review of qualitative literature” is a very good manuscript with the innovative aspects of vaccination studied by the author qualitatively. The author conducted a detailed study and discuss the literature accordingly. However, the author should address the following issues in the manuscript before to accept for publication

  1. The author should take care of the title; it seems the first-word ‘Title” is not the actual title part. It may be from the template. So need to correct it
  2. Box 1 is complex and difficult to understand, it is better to present it in a simple form like a scheme or figure or table, or flow chart form
  3. In figure1, the direction of the arrows downwards are not straight, suggested to make them straight
  4. Table 1 caption is not clear very difficult to understand
  5. In table 2, studies columns, 1 to 32 details are not in the same style. It should start at the same place. Please align them properly
  6. The conclusion need to improve further and include the major outcome of this study
  7. In section 6. Patents, there is no information, so you can delete it
  8. The author can improve the figure 1 quality
  9. Overview of the study as a scheme needs to be incorporated

Author Response

The manuscript by Natacha et al. “Nurses’ influenza vaccination and hesitancy A review of qualitative literature” is a very good manuscript with the innovative aspects of vaccination studied by the author qualitatively. The author conducted a detailed study and discuss the literature accordingly. However, the author should address the following issues in the manuscript before to accept for publication. Thank you for your general comment and the interest in our work.

  1. The author should take care of the title; it seems the first-word ‘Title” is not the actual title part. It may be from the template. So need to correct it thank you: corrected
  2.  
  3. Box 1 is complex and difficult to understand, it is better to present it in a simple form like a scheme or figure or table, or flow chart form thank you for the remark, we agree that it is difficult to read, but its publication ad integrum allows us to redo the study if necessary
  4. In figure1, the direction of the arrows downwards are not straight, suggested to make them straight thank you: corrected
  5. Table 1 caption is not clear very difficult to understand thank you: thank you we have rewritten the title
  6. In table 2, studies columns, 1 to 32 details are not in the same style. It should start at the same place. Please align them properly thank you: corrected
  7. The conclusion need to improve further and include the major outcome of this study thank you: the conclusion has been rewritten
  8. In section 6. Patents, there is no information, so you can delete it thank you: corrected
  9. The author can improve the figure 1 quality thank you: corrected
  10. Overview of the study as a scheme needs to be incorporated :thank you: we made a mindmap: https://miro.com/app/board/uXjVO2We_pU=/ or https://miro.com/app/live-embed/uXjVO2We_pU=/?moveToViewport=-177,-539,1607,850

Round 2

Reviewer 1 Report

Failed to revise according to previous comments

Author Response

Dear Reviewer,

As requested, we have extended our systematic review to the web of science database.

We hope that you will be satisfied with this work.

Please find the new manuscript enclosed.

Sincerely yours,

Round 3

Reviewer 1 Report

We have no more questions about the manuscript

Author Response

Dear editor,
Thank you for your proofreading, systematic has been added in the title and we have specified 2 questions PICO a priori.
Hoping that it is what you expected.

Sincerely,